# Pelagic fish predation is stronger at temperate latitudes than near the equator

Marius Roesti [1,2,3✉], Daniel N. Anstett[1,4,10], Benjamin G. Freeman [1,2,10], Julie A. Lee-Yaw[1,4,5,10], Dolph Schluter[1,2,10], Louise Chavarie[1,2,6], Jonathan Rolland[1,2,7] & Roi Holzman [1,2,8,9]

Species interactions are widely thought to be strongest in the tropics, potentially contributing to the greater number of species at lower latitudes. Yet, empirical tests of this "biotic interactions" hypothesis remain limited and often provide mixed results. Here, we analyze 55 years of catch per unit effort data from pelagic longline fisheries to estimate the strength of predation exerted by large predatory fish in the world's oceans. We test two central tenets of the biotic interactions hypothesis: that predation is (1) strongest near the equator, and (2) positively correlated with species richness. Counter to these predictions, we find that predation is (1) strongest in or near the temperate zone and (2) negatively correlated with oceanic fish species richness. These patterns suggest that, at least for pelagic fish predation, common assumptions about the latitudinal distribution of species interactions do not apply, thereby challenging a leading explanation for the latitudinal gradient in species diversity.

[1] Biodiversity Research Centre, University of British Columbia, Vancouver V6T 1Z4, Canada. [2] Department of Zoology, University of British Columbia, Vancouver V6T 1Z4, Canada. [3] Institute of Ecology and Evolution, University of Bern, 3012 Bern, Switzerland. [4] Department of Botany, University of British Columbia, Vancouver V6T 1Z4, Canada. [5] Biological Sciences, University of Lethbridge, Lethbridge, AB T1K 3M4, Canada. [6] Scottish Centre for Ecology and the Natural Environment, Institute of Biodiversity, Animal Health & Comparative Medicine, University of Glasgow, Glasgow, UK. [7] Department of Computational Biology, University of Lausanne, Quartier Sorge, 1015 Lausanne, Switzerland. [8] School of Zoology, Tel Aviv University, 6997801 Ramat Aviv, Israel. [9] Inter-University Institute for Marine Sciences, 8810302 Eilat, Israel. [10]These authors contributed equally: Daniel N. Anstett, Benjamin G. Freeman, Julie A. Lee-Yaw, Dolph Schluter. ✉email: marius.roesti@iee.unibe.ch

Since Darwin and Wallace, biologists have suggested that interactions between species become stronger towards the equator[1–5]. The prevailing idea is that less extreme climatic conditions in the tropics allow for an increase in the frequency and intensity of species interactions, such that species interactions become the dominant selective force at lower latitudes[2,4,6,7]. The strong selection generated by species interactions may in turn spur faster evolutionary change and potentially speciation at lower latitudes[8], explaining in part why there are more species near the equator than in temperate or polar regions on Earth[8–11] (but see refs. [12,13]). Many studies have evaluated this "biotic interactions" hypothesis (sensu Schemske[9]) by attempting to quantify latitudinal variation in the strength of species interactions. However, most of these studies have been limited in scale and have yielded mixed results, such that the generality of the biotic interactions hypothesis remains difficult to evaluate and controversial[5–7,14–21].

In this study, we evaluate latitudinal patterns of predation in the world's oceans. Predation is a ubiquitous and ecologically important species interaction[22,23] that is known to promote diversification in various organisms[24–28]. Several studies have measured predation at different latitudes in various systems to test predictions arising from the biotic interactions hypothesis. The evidence that predation is strongest near the equator is mixed. For example, some studies have reported an increase in predation on bird eggs and nestlings[5,29,30], insects[17,31], marine invertebrates[32–34], and seeds[21] towards the equator. Other studies, however, have found no relationship between bird nest predation[35] or marine herbivory[4,19] and latitude, or have reported strongest predation on brachiopods at temperate latitudes[18]. Even the fossil record provides inconsistent findings, with strongest historical predation on certain gastropod species in the tropics[36], but a lack of any consistent latitudinal pattern in brachiopod defenses that would signal a trend in predation[37]. Synthesizing this conflicting literature is difficult because individual studies use different methodologies to measure predation and are often limited to relatively small spatial and temporal scales[5,20,38]. Hence, global patterns of predation across space and time remain unclear.

Here we draw upon four massive datasets from 55 years of pelagic longline fishing in the world's major oceans to test latitudinal variation in predation. Specifically, we quantify the relative predation pressure exerted by large fish predators on their prey in the open ocean (hereafter simply 'relative predation'). In contrast to the biotic interactions hypothesis, we find (1) a globally and temporally consistent pattern of stronger relative predation away from the equator and (2) that predation strength is negatively associated with open-water fish species richness.

## Results and discussion

**Relative predation is strongest in or near the temperate zone.** Dividing each of the East and West Pacific, Atlantic, and Indian Oceans into a 5° × 5° grid, we calculated 42,050 annual estimates of relative predation using nominal catch-per-unit-effort (the number of fish predators caught per hook set) from over 900 million attacks by large fish predators (e.g., tunas, billfish, sharks) on hooks baited with natural prey species (e.g., mackerels, herrings, sardines). Data spanned 55 years between 1960 and 2014 and a maximal latitudinal range between 50°S and 60°N (Supplementary Tables 1–3). We found substantial latitudinal variation in relative predation in all four ocean basins (Fig. 1a, Supplementary Table 4). However, relative predation was in no case strongest near the equator as expected under the biotic interactions hypothesis. Instead, relative predation peaked in or near the temperate zone (Fig. 1b). This pattern was particularly

pronounced in the Southern hemisphere, where relative predation was ~65% stronger on average in the temperate zone (latitudes 30°S to 40°S) than near the equator (5°S to 5°N). In the Northern hemisphere, relative predation was ~23% stronger in the temperate zone (latitudes 30°N to 40°N) than near the equator. These patterns were robust to how relative predation was calculated (Supplementary Note 1). In both hemispheres, relative predation declined at latitudes beyond 40° toward the poles, presumably associated with a shift in the dominant predator community in the oceans from pelagic predatory fish to non-pelagic (benthic and demersal) fish predators (e.g., flatfish and gadoids) and non-fish predators (e.g., marine mammals and seabirds)[39,40]. The lower relative predation by pelagic predatory fish near the equator when compared to temperate regions thus refutes the idea that predation in the open ocean is strongest in the tropics.

**Patterns of relative predation have been consistent over recent time and are robust.** We explored whether decades of industrial fishing might have altered latitudinal patterns of relative fish predation, potentially driving the observed patterns. Indeed, overall relative predation declined by two- to three-fold in all ocean basins over the 55 years studied (Supplementary Fig. 1, Supplementary Table 4), presumably reflecting the removal of pelagic predatory fish by industrial fishing[41]. Remarkably, these declines do not seem to have altered the general pattern of relatively stronger fish predation at temperate latitudes than near the equator. Examination of different time slices of the data revealed that the overall latitudinal pattern observed was already evident at the start of our time-series in the early 1960s (Fig. 2, Supplementary Fig. 2), when longline fishing (Supplementary Fig. 3) and industrial fishing in general were much less intense and harvesting impacts still relatively minor[42,43]. Similarly, the stronger fishing pressure and greater oceanic exploitation in the Northern hemisphere[43] is unlikely to be the sole explanation of why relative predation is generally stronger in the Southern hemisphere because this difference between hemispheres was already evident in the early 1960s (Fig. 2). Although we are unable to infer patterns of pelagic fish predation in deep evolutionary time or prior to any human impact on the ocean, the relative stability of the observed latitudinal pattern in more recent time—despite an increase in human fishing pressure—suggests that peaks in pelagic fish predation away from the equator may have been the prevailing pattern over time.

Because global fisheries data are not collected with the specific purpose of measuring differences in predation across latitude, we investigated several factors that could possibly differ across latitude and thus explain our results: seasonality in (i) fishing pressure (Supplementary Figs. 4 and 5) and (ii) the latitudinal pattern of relative predation[44] (Supplementary Fig. 6); (iii) variation in the number of predatory fish taxa reported (Supplementary Fig. 7) or (iv) between targeted and non-targeted taxa by longline fisheries (Supplementary Figs. 8–11); (v) saturation of hooks (Supplementary Fig. 12); (vi) heterogeneity in sampling effort (Supplementary Fig. 13); and (vii) potential errors in data reporting (Supplementary Fig. 14). Overall, we found that these factors do not explain why we found relative predation to be stronger away from the equator. We further examined predation peaks of individual predator taxa and found that, across all ocean basins, only 14% of all taxa demonstrate peak predation rates near the equator (i.e., between latitudes 10°S and 10°N) (Fig. 3). Hence, the general pattern of strongest pelagic fish predation away from the equator appears to hold even when considering individual predator taxa. Finally, our results are robust to analyses that account for variation in other spatial variables, including longitude, proximity to land, and ocean depth

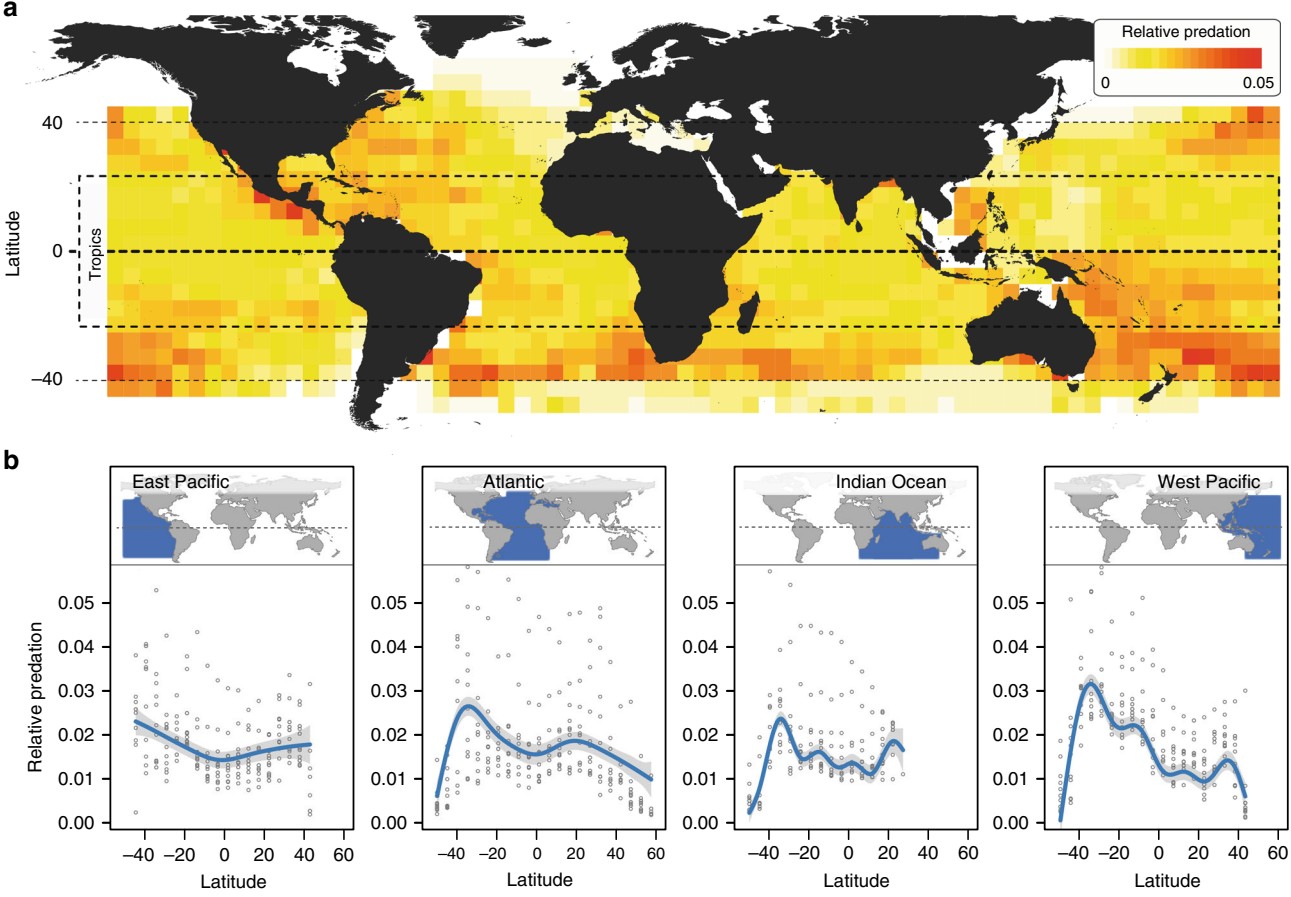

**Fig. 1 Latitudinal variation in relative predation by top fish predators in the open ocean based on pelagic longline fishing data. a** Global map depicting median annual relative predation between 1960 and 2014 at 5° × 5° resolution. The tropics are defined as the region between latitudes 23.5°S and 23.5°N. **b** Partial effect of latitude on variation in relative predation in a generalized additive mixed-effect model (GAMM) run separately for each of four ocean basins (P-values for the partial effect of latitude are below 0.0001 in all four GAMMs; see Supplementary Table 4 for details). This analysis accounted for the effects of both time and spatial autocorrelation in the data (see "Methods" section for details). Blue lines depict the GAMM-predicted function with 95% confidence intervals (gray shading). Gray circles indicate median relative predation per latitude within 5-year time intervals. Source data are provided in Supplementary Data 1.

(Supplementary Fig. 15). We conclude that although some variation in relative predation is explained by how, where, and when the underlying data were collected, we found no evidence or have little a priori reason to believe (considering factors we were unable to directly address due to missing information, such as soak time, hook depth, and bait type) that this variation does more than add noise to a robust latitudinal pattern in the relative strength of predation exerted by large pelagic fish.

**Relative predation and species richness are negatively correlated.** Perhaps the most provocative part of the species interactions hypothesis is that strong ecological interactions occurring over short time scales lead to increased species richness over evolutionary time, either by driving speciation or by facilitating the coexistence of species[9,45]. In contrast to this idea, we found a negative association between the strength of relative predation and overall oceanic fish species richness (calculated using data from *AquaMaps*[46]; see "Methods" for details) across latitude in three out of four ocean basins (Fig. 4). While species richness of open ocean fish in our analysis shows the prototypical peak near the equator, we note that species richness in some other groups of marine organisms peaks away from the equator[47–50]. Future studies should investigate the strength of species interactions in these other groups to test the generality of our finding that species

richness is not positively correlated with the strength of species interactions in the ocean. Interestingly, however, our finding that species interactions are strongest away from the equator aligns with the finding that recent speciation rates in marine fish increase from the equator towards the poles[51], potentially supporting the hypothesis that stronger biotic interactions are indeed associated with faster diversification. Evolutionary ecologists are now charged with explaining why the eco-evolutionary processes thought to generate and maintain diversity are not always strongest in the most diverse regions on Earth.

## Methods
**Data and filtering.** We quantified the relative predation that is exerted by large pelagic fish predators based on four publicly available datasets from pelagic longline fishing (West Pacific, East Pacific, Atlantic, and Indian Ocean), each managed by an independent commission (see Supplementary Table 1 for details). All datasets included catch per effort records per month and year at spatial resolutions of 1° × 1° or 5° × 5° latitudinal grids (a small percentage of records reported at lower resolution were removed). For consistency, we aggregated data presented at 1° × 1° resolution to the lower resolution by assigning records to the nearest geographic midpoint of an overlaid 5° × 5° grid. Data availability was limited prior to 1960, and we thus considered only records from 1960 onward (up to 2014 to permit 5-year blocking of the data; see below).

We filtered our datasets in several ways. We filtered out catch per effort records from longlines that were set to catch a specific target species. We also removed incomplete records (when either catch or effort was not reported), as well as suspect entries for which (i) the total number of predators caught was greater than

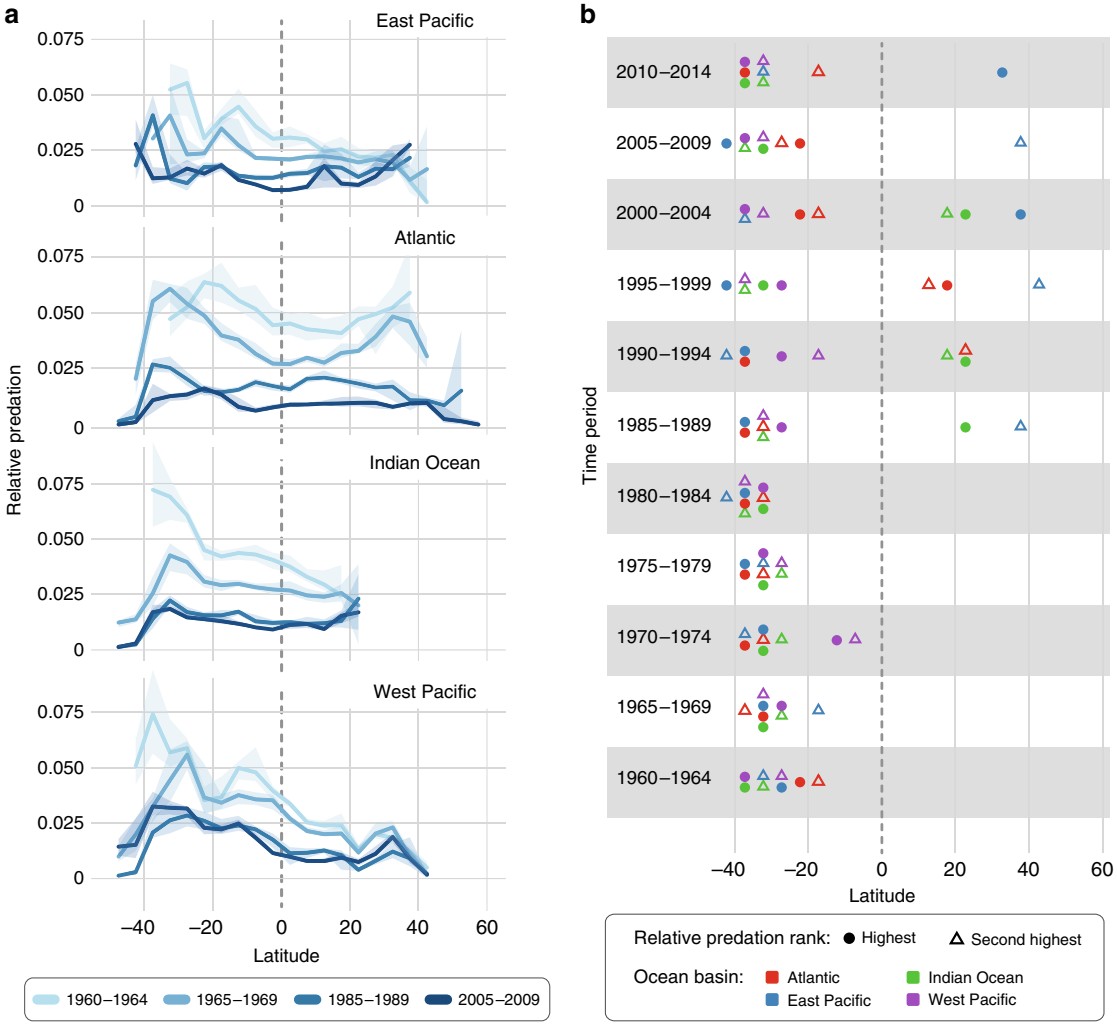

**Fig. 2 Temporal patterns of relative predation across latitude. a** Median annual relative predation (with non-parametric 95% confidence intervals) exerted by pelagic fish predators across latitude for four representative 5-year time intervals. **b** Latitudes with the strongest (dots) and second strongest (triangles) median relative predation per 5-year time interval between 1960 and 2014. Source data are provided in Supplementary Data 1.

the actual number of hooks set; (ii) the number of hooks set or the number predators caught was not a whole number; or (iii) the number of predators caught was recorded as zero (zero catches are highly unlikely given the many thousands of baited hooks that are deployed into the ocean on each longline; records with zero catches represented no more than 2% of the data in any ocean basin). After filtering (and later removing of extreme [outlier] values, see below), we obtained a total of 359,584 catch per effort records from the 55-year period between 1960 and 2014 across all ocean basins (Supplementary Table 2). The maximal latitudinal range covered by the data was 50°S to 60°N (Supplementary Table 2). We consider the tropics to be the region between latitudes 23.5°S and 23.5°N.

**Quantifying predation**. We estimated annual relative predation for each grid cell as the total number of fish predators caught in a year divided by the total number of hooks set that year (akin to a nominal estimate of catch-per-unit-effort, CPUE). This way of measuring predation experienced by a prey organism placed into the wild follows the logic of previous experimental tests assessing predation strength (e.g., refs. [17,30]; see ref. [52] for a review). We note that there has been some debate on biases in such tethering experiments, where prey do not have an ability to escape[53,54], potentially leading to an overestimation of predation[54]. However, the size and scope of our datasets may buffer against some of these potential biases, and an overestimation of predation is unlikely to explain relative differences in predation strength across locations to which the same methodology was applied.

Our study is focused on quantifying predation by a large number of open-ocean predatory fish species including tuna, billfish, and shark species (see Supplementary Table 3), but it does not quantify predation by non-pelagic fish predators or by non-fish predators. Acknowledging that we therefore do not quantify total oceanic predation, we refer to the metric of predation used as relative predation. Importantly, however, pelagic fish dominate the predator community between latitudes of about 40°S to 40°N[39], which encompasses the majority of our

study area. Thus our results are expected to represent a large percentage of the total predation experienced in the study area.

**Patterns of variation in predation**. We analyzed the relationship between latitude and relative predation separately for each ocean basin using generalized additive mixed-effect models (GAMMs). We used GAMMs because these models infer the simplest function describing variation in relative predation across latitude without making any a priori assumptions about the form of this relationship (e.g., linear, quadratic, etc.). We accounted for spatial autocorrelation in the data by including a spherical data correlation structure term in the model[55]. The latitude and longitude of each grid cell were re-projected to ocean-specific equidistant projections prior to calculating the correlation structure. To reduce possible effects of temporal auto-correlation in our data, we calculated median annual relative predation per grid cell within 5-year time intervals, rather than relying on annual estimates. In addition to 'latitude', we included 'time interval' as a fixed effect to examine and control for changes in overall relative predation through time. 'Grid cell' was included as a random variable to account for the repeated nature of the data. GAMMs were conducted using thin plate regression splines (specified by *bs* = '*tp*' in the model syntax, see below) as implemented in the *mgcv*[56] R-package. The final GAMM syntax was

*gamm(attack.rate ~ s(latitude, bs = "tp") + s(time interval, bs = "tp"), random = list(GridID = ~1), correlation = corSpher(form = ~ latitude_ed + longitude_ed), data = data).*

**Robustness checks and accounting for methodological variation**. We tested the robustness of the overall latitudinal pattern of relative predation against the influence of several potential confounding factors.

Seasonality in fishing pressure: We tested whether seasonality in fishing pressure varied systematically across latitude, potentially biasing the data. We

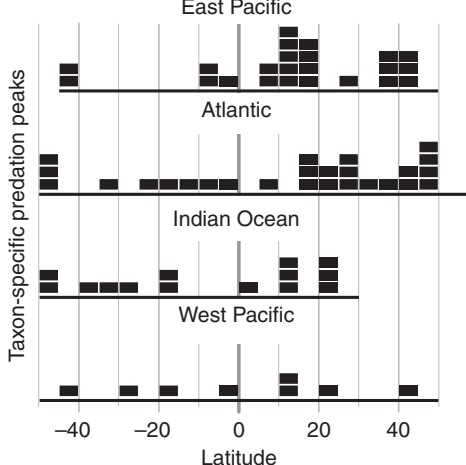

**Fig. 3 Distribution of taxon-specific predation peaks across latitude.** The latitude where the predation exerted by each predator taxon (i.e., species or groups of related fish species) is strongest was calculated based on yearly catch per effort data, separately for each ocean basin. Each black box indicates the latitudinal position of one taxon-specific predation peak (number of taxa: East Pacific N = 23, Atlantic N = 26, Indian Ocean N = 14, West Pacific N = 8; See Supplementary Fig. 9 and Supplementary Table 3 for detailed taxonomic information). The length of the x-axis indicates the maximal latitudinal extent covered by the available data for each ocean basin. Source data are provided in Supplementary Data 1.

re-ran our ocean-specific GAMMs accounting for the partial effect of season on relative predation, by adding 'month' as an additional fixed effect (along with 'latitude' and 'time interval') into the models. While we do see some evidence of seasonal variation in fishing pressure (Supplementary Fig. 5) and in the latitudinal pattern of relative predation (Supplementary Fig. 6), this variation does not influence our main finding of stronger relative predation away from the equator (Supplementary Figs. 4 and 6).

Number of predatory fish taxa reported and targeted by longline fisheries: We evaluated whether the number of reported fish predator taxa was higher at those latitudes where relative predation was strongest (Supplementary Fig. 7; note that we use the terminology of predator 'taxon' instead of 'species' throughout this study because longline fisheries catches are not always provided per predator species, but are sometimes aggregated for a group of related fish predator species; see Supplementary Table 3 for details). We then tested whether the bias of fisheries towards certain target species (such as Bigeye or Albacore tuna)—thus resulting in an underrepresentation of non-targeted species—affects latitudinal patterns of relative fish predation based on longline data. Although we could not fully account for missing predators, we asked whether the latitudinal patterns observed for fish predators that are not major targets of longline fisheries are consistent with the patterns observed based on the full datasets. Specifically, we re-ran the GAMMs using estimates of relative predation based only on predator taxa that were deemed 'non-target' fish of pelagic longline fisheries (i.e., taxa making up <10% of the total catch within each ocean; Supplementary Table 3). Although temperate peaks in relative predation are weaker in this analysis, we still fail to find evidence to suggest that relative predation is strongest at the equator (Supplementary Fig. 8). Finally, we determined the latitude with the strongest predation per fish predator taxon based on latitudinal means of annual total catch of every fish predator taxon, divided by the total effort per grid cell. Despite marked taxon-specific variation in the strength of predation across latitude, most predation peaks fall away from the equator (Fig. 3 and Supplementary Fig. 9; latitudinal predation patterns for every taxon are depicted in Supplementary Figs. 10 and 11).

Saturation of hooks and heterogeneity in sampling effort: We investigated whether observed patterns of relative predation across latitude could be biased by hook saturation, whereby an increasing number of hooks set would lead to

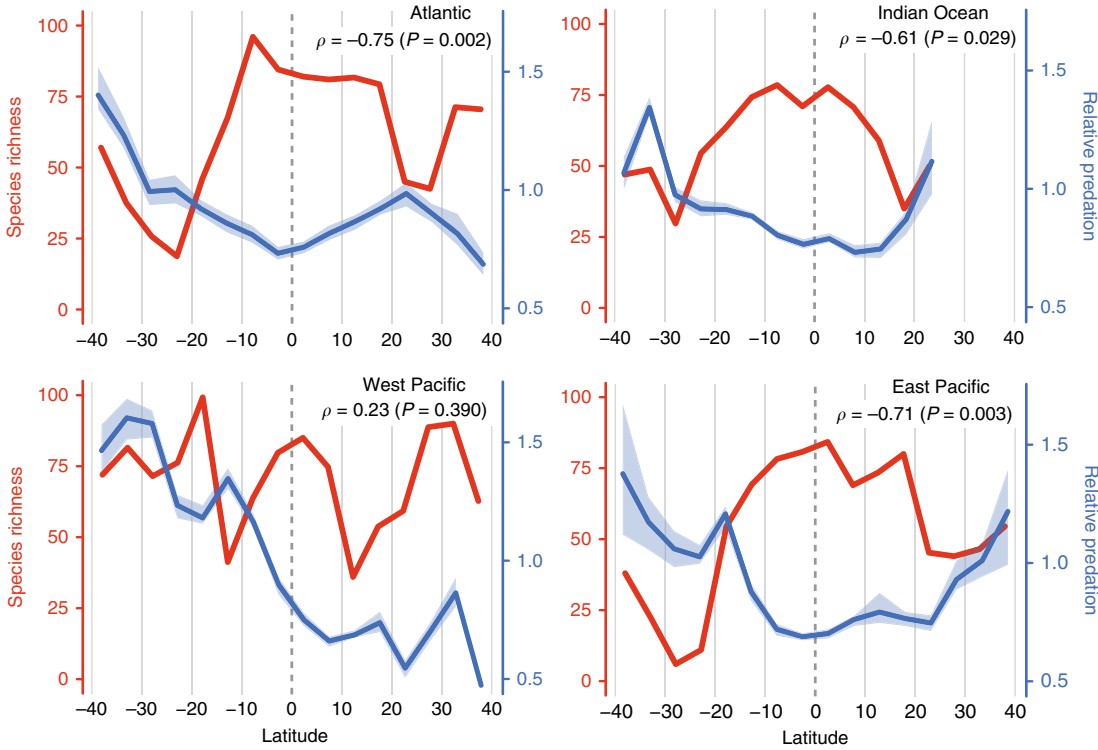

**Fig. 4 Association between latitudinal variation in fish species richness and relative predation.** Median number of pelagic fish species was estimated from global species range maps[46] (red lines), and compared to median relative predation by predatory fish across latitude in bands of five degrees (blue lines with non-parametric 95% confidence intervals). To account for the effect of time in overall relative predation (see Supplementary Fig. 1), annual relative predation estimates were standardized by the mean relative predation of the respective year prior to calculating latitudinal medians. Spearman's rho and the statistical significance (non-parametric P-values) of the association between median species richness and median relative predation across latitude are given in the top right of each plot. Because non-pelagic fish and non-fish predators are the dominant oceanic predators at absolute latitudes >40°[39,40], this analysis was restricted to latitudes between 40°S and 40°N. Source data are provided in Supplementary Data 1.

diminishing catch returns. This effect might be exacerbated by removal of fish predators by the fishery itself over time, resulting in lower estimates of relative predation where the number of hooks set (effort) has been excessive. We tested for a saturating relationship between catch and effort in each grid cell within consecutive time periods of 11 years (i.e., 1960–1970, 1971–1981, etc.) using the 359,584 individual catch per effort data records (Supplementary Table 2). Focusing on 11-year time periods provided sufficient individual data records per grid cell to test for hook saturation. Specifically, for each geographic grid cell and time period with at least 20 individual data records, we fitted the exponential function:

$$y = ax^b,$$

where $y$ represents catch and $x$ represents effort (number of hooks). We then took the logarithm of both sides of the formula to fit the linear model:

$$\log(y) = \log(a) + b \times \log(x)$$

A value of $b$ less than one should thus indicate hook saturation in a given location and time period. We used GAMMs to determine the extent to which $b$ varies with latitude. Separate models were run for each ocean basin, with 'latitude' as a fixed effect, 'grid cell' as a random effect, and a correlation term accounting for spatial autocorrelation (akin to the GAMM syntax described in the previous section). Because models including 'time period' as a fixed effect did not converge, we accounted for the effect of time by standardizing each slope value by the mean slope across all grid cells from the respective time period. We found that the majority of estimates of $b$ were above 1 rather than below or equal to 1, indicating higher catch per effort with increasing effort rather than saturation (Supplementary Fig. 12A). We lack an explanation for this trend. However, critically, $b$ did not vary greatly with latitude, and consequently did not show the latitudinal variation observed in relative predation (compare Supplementary Fig. 12B with Fig. 1b). Moreover, there was either no or a weak but inconsistent association between total effort and relative predation across latitude in the oceans (Supplementary Fig. 13). The overall pattern of relative predation across latitude is therefore unlikely to be explained by saturation and differences in fishing efforts.

Potential errors in data reporting: GAMMs may be sensitive to extreme values, so we ran GAMMs both with and without values of relative predation that were deemed high or low outliers. Because predation strength was expected to vary by latitude and time, outlier values were identified separately for each latitude in 5-year time intervals based on the individual catch per effort data prior to annual pooling. A value was considered an outlier if it was above or below four standard deviations of the median relative predation within a given latitude–time interval subset. Outlier values represented no more than 2% of the data in any dataset and their removal did not qualitatively affect the results (Supplementary Fig. 14). Our study generally reports results based on datasets with outlier values removed.

Spatial variation in addition to latitude: We re-evaluated the latitudinal pattern of relative predation while accounting for a possible influence of other geographical variables—including longitude, ocean depth, and distance to land—on relative predation. Although some additional variation of relative predation is explained by these variables, the main latitudinal pattern of relative predation remains unchanged (Supplementary Fig. 15).

Further methodological details on all robustness checks are provided as part of the respective figure legends in the Supplementary Material.

**Association between latitudinal variation in predation and species richness.** We obtained estimates of oceanic fish species richness from *AquaMaps*[46]. In line with our focus on open-ocean (pelagic) fish predation, we restricted estimates of species richness to "pelagic-oceanic" fish (see *FishBase*[57] for classification). Species richness estimates were originally available at 0.5° × 0.5° resolution. To aggregate estimates to the same 5° × 5° resolution used to analyze relative predation, we first consolidated the original records into 1° × 1° grid cells, calculating the mean value for each new cell (four records per cell). We then overlaid our 5° × 5° grid on this grid and assigned species richness to the final grid based on the value at the midpoint of each 5° × 5° cell (where the midpoint of a 5° × 5° grid cell did not correspond to a single estimate of species richness, we averaged across all estimates from the 1° × 1° grid adjacent to the midpoint). To examine the relationship between latitudinal variation in pelagic fish species richness and predation, we calculated median species richness and median annual relative predation for each 5° of latitude from 40°S to 40°N (to account for the effect of time in the overall relative predation [Supplementary Fig. 1], annual relative predation estimates were standardized by the mean value across all cells for that year prior to calculating the median). Spearman's rank correlation (Rho) was used to quantify the relationship between median species richness and median relative predation for these 5° increments of latitude. We note that the species richness estimated by *AquaMaps*[46] is based on range maps, potentially leading to richness overestimates due to errors of commission. However, the latitudinal species richness pattern we observe is likely to be robust to this error because this and related reporting errors are unlikely to vary systematically across latitude.

**Reporting summary.** Further information on research design is available in the Nature Research Reporting Summary linked to this article.

## Data availability

Source links to obtain the publicly available raw longline fisheries datasets here analyzed are provided in Supplementary Table 1. The source data (after filtering out outliers; see "Methods" section) underlying Figs. 1–4, Supplementary Table 3 and Supplementary Figs. 1, 2, 7–13, 15, 19–22 are provided in Supplementary Data 1.

## Code availability

Scripts used for initial processing and filtering of the raw longline data (c_DataPrep_allOceans.r), to add ocean depth (c_Ocean.depth.r) and distance to land (c_Distance.to.land.r) to the longline data, and to obtain and visualize the results shown in Fig. 1 (c_Main.Analysis.r) are available as Supplementary Software 1. All analyses and plotting for this work were done in R[58].

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

## Acknowledgements

We thank Peter Williams, Paul Taconet, Carlos Palma, and Doug Bear for help obtaining the fisheries datasets, and Kristin Kaschner, Rainer Froese, Kathleen Reyes, and Cristina Garilao for providing the fish species richness data from *AquaMaps*. This study benefited from input by Darren Irwin. M.R. thanks Katie Peichel for letting him work on this study while being part of her group at the University of Bern. M.R. was funded by postdoctoral fellowships of the Swiss National Science Foundation (P2BSP3_161931, P300PA_174344); D.N.A. was funded by Banting and NSERC postdoctoral fellowships and a Killam fellowship; J.A.L-Y. was funded by a NSERC postdoctoral fellowship (#487035); B.G.F. was funded by postdoctoral fellowships from the UBC Biodiversity Research Centre and Banting (#379958); L.C. was funded by the W. Garfield Weston Foundation and the Great Lakes Fisheries Commission; J.R. was funded by a Marie Skłodowska-Curie grant (#785910) and a Banting postdoctoral fellowship (#151042). The Swiss National Science Foundation financially supported the open access publication of this study.

## Author contributions

M.R. had the idea for this project, which was then followed up on by R.H. and M.R.; M.R. and R.H. conducted the initial (main) analyses with important conceptual input by D.S. and J.A.L-Y.; M.R. visualized the results, conducted the analyses required for revising the manuscript and oversaw the project; M.R., D.N.A., B.G.F., and J.A.L-Y. all wrote the paper. All authors (M.R., D.N.A., B.G.F., J.A.L-Y., D.S., L.C., J.R., R.H.) contributed to some degree to the design of the analyses, interpretation of results, and commented on the final manuscript. All authors approved the final manuscript.

## Competing interests

The authors declare no competing interests.
