## [Peer Review File · Nature Communications]

Reviewers' Comments:

Reviewer #1:

Remarks to the Author:

I was excited to read this paper, and at first, it did not disappoint. It was well written. There was a global data set and impressive statistical analysis suggesting a pattern contrary to expectations, namely that predation pressure is higher in mid than low latitudes, even though there are more species in low latitudes.

However, buried away at the back of the paper I discover there were no data on actual predator attack rates. It is only CPUE. There is no evidence provided that this reflects predator attack rates. I see why it might, but other factors may also relate to it. For example, water transparency (fish may avoid hooks if they see them attached to lines), length of time hooks were in the water (called soak time), different species or body sizes in different places, sea temperature during the fishing time, depth of hooks. We are not told exactly what species are where, and they probably vary in abundance with latitude. It may be one thing to show these different CPUE and suggest they may reflect relative predation pressure, but to pretend actual predation was measured is misleading. The lack of mention of the true nature of the underlying data on page 2 seems designed to mislead the readers.

The paper does not adequately review the compelling evidence for greater predation pressure in the tropics (e.g., see classic studies by Geerat Vermeij). Also, competition is never mentioned as a factor in speciation. A recent study by Graham Edgar and others (in this journal or perhaps *Science Advances*) found a contrasting latitudinal gradient in fish and invertebrates and suggested this may be due to competition.

To understand biodiversity gradients like this, it is necessary to consider the species involved and their ecology. However, we are not told what the species are nor how their relative abundance varied in space and time. This is missing from the SM as well.

Were longitudinal or other patterns search for in the data (apart from years)? Looking across data in Figure 1 suggests there is also longitudinal variation. So what does that mean and how did it arise?

It is well known that there are other mechanisms in speciation pressure (line 95), indeed these are more plausible and convincing than predation already (e.g., vicariance, temperature related body size, mutations and generation time).

It is interesting that all latitudinal gradient curves dip near the equator, as suggested is the case for marine species by Chaudhary et al. (*TREE* 2017 & 2018). The gradients also seem similar over time periods which is interesting, but decreasing over time, presumably due to over-fishing and stock declines. These patterns merit further study and perhaps publication.

Reviewer #2:

Remarks to the Author:

The manuscript by M. Roesti and coauthors aims to test whether (a) species interaction strength varies across latitudes for pelagic fish in open ocean across multiple ocean basins and (b) this corresponds to species richness for pelagic fish. They analyze an extensive data set from longline fishing catch records across a 55 year period, using catch-per-unit-effort (# of large pelagic fish caught / # of "hooks" set with bait fish) as an estimate of predation rate. The CPUE is then estimated as a median value for a five year period in each 5x5 degree grid cell of ocean to test latitudinal pattern

of CPUE as proxy for predation rate, and subsequently compared to independent estimate of species richness, for each of several ocean basins.

The results of these analyses do not support a higher CPUE rate in the tropics compared to temperate latitudes, and they show a negative relationship between this measure of predation rate and species richness of pelagic fish, contrary to prediction.

The approach is novel and of general (high) interest, using an impressive data set on longline fishing records. The manuscript is well-written and presentation of results is very clear. There are however several assumptions in the data and analyses, which require further consideration and scrutiny, before I would find the results robust and convincing. I suggest the following points be addressed:

1. As the authors point out, the data are reported catch of large pelagic fish on bait. Are there regional differences in bait or other methods that affect probability of catch? In some ways, disappearance of bait is equivalent to a "predation" event than catch per se. In many ways, the data are similar to those collected in "tethering experiments" that use live or dead prey, and either examine prey loss or predator catch, and there's always a question about how well these experiments estimate (approximate) actual predator-prey behavior, including a rich literature on this debate. I understand that catch data is what is available, but there should be some further discussion about the potential for bias and implications of both issues in interpretation (use) of these data as proxy for predation intensity.

2. Since they use CPUE as a proxy, what is known about the quality of reporting? I can imagine that only certain types of fish may be of interest (to those fishing) and others potentially discarded as bycatch. To consider "catch" as a measure of predation intensity, it seems important to include all species of fish caught (= total predation) versus only those of commercial value. I don't know much about this industry and whether there is a substantial portion of the catch this is not recorded. This should be addressed, to understand the data quality.

3. I have some concerns and questions about the effect of temporal variation on latitudinal pattern. The authors do a good job of exploring temporal pattern across 5 and 10 year time scales (Fig 2 and S10) to show consistent data at this time block, but I wonder whether the latitudinal pattern observed could be an artefact of pooling into five-year time bins and then looking at median values. I would expect considerable temporal variation and that this varies across latitude. In particular, in some marine systems (eg., benthic communities), predation can be very intense in temperate latitudes during the summer months, when it can be as high as or even higher than tropical latitudes. However, it may be that predation is more consistent in tropical systems for much of the year. If there is strong seasonality in temperature vs tropical latitudes, when predation is measured could greatly affect the latitudinal pattern observed (e.g., see discussion on recent paper by Cheng, Diversity & Distributions 2019, vol. 25). In this study, I would like to know whether there are seasonal differences in the catch data, such that (a) only certain seasons are represented in some latitudes (x ocean) and (b) whether season of measurement effects the results. For example, could higher catch rates in the temperate zone be the result of fishing only during the "high season" compared to tropical areas? Median values or other metrics would be affected by this. It appears that the authors have the catch data by month, so I believe there are opportunities to test whether there are seasonal differences in both when data are collected and catch rates across latitude. While I don't mean that every aspect of this must be explored, it would provide more confidence to know that we aren't just seeing a difference in seasonality in effort or catch, and I can imagine various ways to approach this; one might be to compare summer and winter seasons separately to test whether results are robust to seasonality.

4. I am also struck by several patterns in looking at the spatial display of data in Fig 1 that pose a few

further questions and recommendations:

(a) First, it appears that the patterns adjacent to the continents (e Pacific and w Pacific) does display the predicted latitudinal pattern in the in the northern hemisphere. This is lost/swamped when combining with mid ocean. Is there a difference between continental shelf/adjacent and mid ocean, or with distance from shore x latitude? If so, this is interesting and worth discussion. More broadly, it would be good to discuss briefly what are the expectation (and past literature) for open ocean pelagic fish?

(b) Second, I'm curious why the eastern and western Atlantic aren't separated in the analysis in parallel to both sides of the Pacific. It seems like they should be and would show a different pattern that the entire Atlantic. If this is an issue with space, it could be added to supplemental.

(c) Third, it's also interesting how much higher the catch appears in the southern ocean, and how much lower in the NE Atlantic. Does this suggest historical effects (or lack) of fishing pressure, or is this driven by productivity, or some combination. While exploration of this is beyond the scope of the current paper, it would be worth some mention of possible drivers for this (and especially the southern ocean, since this is such a prominent feature) --- in just a sentence.

Reviewers' comments

Reviewer #1 (Remarks to the Author)

I was excited to read this paper, and at first, it did not disappoint. It was well written. There was a global data set and impressive statistical analysis suggesting a pattern contrary to expectations, namely that predation pressure is higher in mid than low latitudes, even though there are more species in low latitudes.

(1) However, buried away at the back of the paper I discover there were no data on actual predator attack rates. It is only CPUE. There is no evidence provided that this reflects predator attack rates. I see why it might, but other factors may also relate to it. For example, water transparency (fish may avoid hooks if they see them attached to lines), length of time hooks were in the water (called soak time), different species or body sizes in different places, sea temperature during the fishing time, depth of hooks. We are not told exactly what species are where, and they probably vary in abundance with latitude. It may be one thing to show these different CPUE and suggest they may reflect relative predation, but to pretend actual predation was measured is misleading. The lack of mention of the true nature of the underlying data on page 2 seems designed to mislead the readers.

We appreciate this critical input by Reviewer 1. For this revision, we now include several new analyses and have added and altered text. In particular, we now provide

- (i) a justification of using CPUE as measure of predation and of our use of (new) terminology in this paper
- (ii) new analyses of factors potentially influencing/biasing our metric of relative predation
- (iii) information on predator taxa, their abundance and their specific patterns of predation across latitude.

While we largely agree with this criticism by Reviewer 1, we clarify that we had no intention of misleading readers in our original submission.

(i) Justification of using CPUE as a measure of predation and use of terminology

We note that our goal was not to measure total predation (as implied by Reviewer 1), but maybe this was not instantly clear in the previous version of our manuscript. We now clarify this by using the term “relative predation” as suggested by the reviewer throughout the manuscript, and by making very explicit why we use the term *relative* predation, and how it was calculated (lines 175-179): *“Our study is focused on quantifying the predation exerted by a large number of open-ocean predatory fish species including tuna, billfish and shark species (see Table S3), but it does not quantify predation by non-pelagic fish predators or by non-fish predators. Acknowledging that we therefore do not quantify total oceanic predation, we refer to the obtained metric of predation as relative predation throughout our study.”*

Importantly, however, we note that in the bulk of our study area (i.e., latitudes between -40° and 40°) pelagic fish predators are the major predator guild (lines 179-181). We are thus assessing predation strength of the main predators in this area.

To make clear already in the title of the manuscript that we are studying 'fish predation' (and not just any predation) of oceanic (pelagic) predatory species, we have changed the title to "*Fish predation in the world's open oceans is stronger at temperate latitudes than near the equator*".

We now clarify in the main text (lines 63-66), as well as in the Methods (lines 164-166), exactly how relative predation was calculated.

Lines 63-66: "We use nominal catch-per-unit-effort (CPUE), defined as the number of pelagic fish predators caught (catch) per hook set (effort), as a measure of the relative predation pressure exerted by large fish predators on small prey species in the open ocean."

Lines 164-166: "We estimated annual relative predation for each grid cell as the total number of fish predators caught in a year divided by the total number of hooks set that year (akin to a nominal estimate of catch-per-unit-effort, CPUE)."

We now also include the following justification for utilizing nominal CPUE as a measure of relative predation in the main text, which includes references to studies that have applied the same or a very similar approach to measure predation strength (lines 166-168): "*This way of measuring predation that is experienced by a prey organism placed into the wild follows the logic of previous experimental tests assessing predation strength (e.g., ^{16, 29}; see⁵¹ for a review).*"

We back up our findings from the analysis of nominal CPUE with alternative calculations of relative predation based on catch-per-unit-effort data (please see the Supplementary Text "Consideration of alternative estimates of predation" for details).

(ii) Factors potentially influencing/biasing our metric of relative predation

Reviewer 1 points to several factors that may influence our metric of relative predation, thus holding the potential to introduce bias into our analyses. We have explored these factors and now include several new analyses that address these concerns.

Overall, we totally agree with Reviewer 1 that a number of factors may influence our metric of relative predation. However, it is important to distinguish between "natural" factors that are already included within latitude (and hence help create observed latitudinal patterns) and methodological factors that may deserve further scrutiny, as they may confound results.

We consider environmental factors that contribute to latitudinal patterns to be "natural" factors. Consider a case where (model) prey specimens are placed at different latitudinal locations on land to measure attack rates (for such a study, see e.g., ref¹). If these models are put out in some geographic locations where fog is more abundant than in other locations (e.g., cloud forests), and if fog influences the perception of the prey models by visual predators, then fog will be related to lower predation. However, while fog would then be an interesting variable to explain observed geographic patterns, it would not be a variable we would want to control for *a priori*. For example, we consider sea temperature and predator communities to be examples of such "natural" environmental variables. For the purposes of our study (focused on latitudinal patterns

of predation), we have chosen to use latitude as the predictor knowing that it includes effects of temperature and of evolutionary history. Also, the 'biotic interactions' hypothesis – which we aim to test with our study – specifically talks about latitudinal patterns.

We agree with Reviewer 1 that methodological factors that vary between different grid cells may be of concern. Importantly, however, any variation in methodology that occurs randomly with respect to latitude will add noise (and decrease power) but will not add systematic bias to analyses that are explicitly concerned with *latitudinal* patterns. Thus, confounding factors are only then of major worry if they vary with latitude. In a new section in the main text (lines 105 - 119), and in an extended Methods section (lines 204 - 272), we now describe several new analyses conducted for this revision in order to evaluate the influence of potentially confounding factors on latitudinal results.

Water turbidity: Although we believe water turbidity is an environmental factor, we have chosen to look into how this variable affects relative predation. To investigate latitudinal patterns in turbidity, we analyzed chlorophyll-a concentration, a common proxy for water turbidity^{5, 6}. We present these results as Supporting Analysis as part of our Response Letter (please see the end of this document). Reassuringly for our main results, after accounting for the effect of the water turbidity proxy, we still see highly similar latitudinal patterns with increased relative predation at temperate latitudes (especially in the southern hemisphere), and lower relative predation near the equator. We would prefer to not include this analysis in the supplementary materials since we would like to concentrate on methodological analyses but can include it if the reviewers or editor feel strongly about this.

Body size: Similarly, we consider that body size of predators primarily constitutes an environmental factor in predation. Body size is potentially a methodological problem if big fish eat more than small fish (likely true) such that CPUE is biased/low in areas with large predators, because the single large predator is killed by the first hook. However, the evidence for strong latitudinal body size clines in marine fish is mixed^{7, 8, 9, 10, 11}, making it unclear if this scenario applies. It has been suggested that body size clines may be altered by fishing pressure^{12, 13}. If so, we would expect the influence of “natural” body size clines on our data to be strongest at early time points and then become weaker at later time points. However, we have data since the 1960's and still see higher predation at temperate latitudes in the earlier data (Fig. 2). Hence, we argue that our data is somewhat robust to changes in body size.

Seasonality: Although climate seasonality is primarily an environmental variable, seasonality in fishing pressure is a methodological variable and may thus be a source for bias. We therefore investigated the effect of seasonal variation by calculating relative predation per month and taking the median over 5-year intervals of every month (Fig. S4). This follows our general approach for how data were aggregated to input into GAMMs.

Considering the partial effect of latitude (controlling for month and 5-year period) still shows the same pattern of increased estimated predation at temperate latitudes and lower predation in the tropics (Fig. S4A). Furthermore, the partial effect of month does

not vary too much across months (Fig. S4B). These analyses suggest that despite some seasonal variation in fishing pressure (Fig. S5), the overall latitudinal pattern of relative predation are not affected by this variation.

Variation in effort: To address whether variation in effort across space could impact nominal CPUE (#predators caught / #hooks set) and hence our estimate of predation, we carried out new analyses on the number of predators caught and included effort, latitude, and time as fixed factors (Supplementary Material lines 69-77, Fig. S16). This complements our previous analyses in which we show that there is no clear and consistent association between median relative predation and total effort across latitude (Fig. S12) and no latitudinal trend in hook saturation (Fig. S11).

Factors we are unable to control for: We now explicitly acknowledge in a new caveats section in the main text (lines 105-119) that we cannot directly control for all factors and give examples of soak time, depth of hooks, and bait type (lines 115-119).

(iii) Information on predator taxa and their abundance

Both reviewers expressed interest in more detailed taxonomic information. We now include the name of each fish predator taxon recorded in the datasets (i.e., predator species or groups of predator species when categories were not narrowed down to the species level) and its relative abundance in the total catch in a new table (Table S3). We also include plots showing taxon-specific latitudinal patterns of relative predation within each ocean basin (Fig. S9, S10). Most importantly, we find that irrespective of the relative frequency of taxa in the total catch, most taxa show peaks in relative predation at temperate latitudes (Fig. S8).

We also now show separate GAMM-based analyses of only non-target species of longline fisheries (defined as any predator taxon that contributes less than 10% to the total catch within an ocean basin) (Fig.S7). We think this is an excellent addition to our manuscript, as it directly compares patterns between the full data set and for the subset of species that are not actively sought after. These new analyses fail to find peaks of strongest relative predation near the equator, supporting our results from the analyses including all species.

Finally, we show in a new Fig. S6 that the resolution of recorded predator taxa is not greater at temperate latitudes than around the equator. This emphasizes that the detected pattern of higher overall relative predation at temperate than equatorial latitudes cannot be an artifact of having a greater predator species resolution in the temperate zone.

(2) The paper does not adequately review the compelling evidence for greater predation pressure in the tropics (e.g., see classic studies by Geerat Vermeij). Also, it is never mentioned as a factor in speciation. A recent study by Graham Edgar and others (in this journal or perhaps Science Advances) found a contrasting latitudinal gradient in fish and invertebrates and suggested this may be due to competition.

We have now added a substantial paragraph that addresses previous work on

latitudinal gradients in predation (lines 47-61). We have reworked our paragraph on species interactions promoting speciation to talk about species interactions in general rather than just predation. Here, we have added two more papers demonstrating a role of predation in diversification (line 49; ref. 26 & 27 of the main paper). We now also cite Edgar et al. 2017 Science Advances (line 123, ref. 44 of the main paper). Finally, we explicitly mention 'competition' as another important species interaction (line 51-52).

(3) To understand biodiversity gradients like this, it is necessary to consider the species involved and their ecology. However, we are not told what the species are nor how their relative abundance varied in space and time. This is missing from the SM as well.

Both reviewers requested further data detailing the predator species involved, and we have added this information to our revised manuscript. Please see our response to comment (1) (subsection iii), and the new Table S3 where all species, as well as their relative abundance, are tabulated.

(4) Were longitudinal or other patterns search for in the data (apart from years)? Looking across data in Figure 1 suggests there is also longitudinal variation. So what does that mean and how did it arise?

This is an interesting observation indeed. To investigate this observation, we ran a new set of GAMMs in which we accounted for longitudinal variation (as well as other spatial variation related to proximity to land and ocean depth) besides latitude and time (Fig. S14). Although there is indeed some longitudinal variation in relative predation present, the latitudinal pattern we observed previously remains largely unchanged after accounting for this longitudinal variation. We now state this finding in the main text (lines 113-115): *"Similarly, we found our results to be robust to analyses accounting for variation related to other spatial variables (longitude, proximity to land and ocean depth; Fig. S14)."*

(5) It is well known that there are other mechanisms in speciation pressure (line 95), indeed these are more plausible and convincing than predation already (e.g., vicariance, temperature related body size, mutations and generation time).

We have reworked our paragraph on the hypothesis that species interactions can possibly promote speciation (lines 120-136; please also see our response to comment 2).

We note that with the analysis shown in Fig. 3 we test one specific (and among some people very popular) prediction of how the intensity of species interactions is related to diversity across latitude. The prediction is that there is a positive relationship. However, we do so without making the claim that biotic interactions are strongly linked to speciation, nor that the biotic interactions hypothesis is any more plausible than any number of other hypotheses predicting species diversity across latitude (most of which are not mutually exclusive). We hope Reviewer 1 understands that a review of the many possible processes involved in explaining this major biogeographic pattern is beyond the scope of this paper and has already been extensively covered in the literature^{14, 15, 16, 17}.

(6) It is interesting that all latitudinal gradient curves dip near the equator, as suggested is the case for marine species by Chaudhary et al. (TREE 2017 & 2018). The gradients also seem similar over time periods which is interesting, but decreasing over time,

presumably due to over-fishing and stock declines. These patterns merit further study and perhaps publication.

Thank you for your interest in our findings. We agree that this pattern merits further study. Indeed, the two papers highlighted here are relevant to our work and they are now cited also with a note on potential future research (lines 126-130). *"While species richness of the open ocean fish we analyzed shows the prototypical peaks at the equator, we note that species richness of some other groups of marine organisms peaks away from the equator"^{46, 47, 48, 49}. Future studies should investigate interaction strength in these other groups to test the generality of our finding that species richness is uncorrelated with interaction strength in the ocean."*

#####

Reviewer #2 (Remarks to the Author)

The manuscript by M. Roesti and coauthors aims to test whether (a) species interaction strength varies across latitudes for pelagic fish in open ocean across multiple ocean basins and (b) this corresponds to species richness for pelagic fish. They analyze an extensive data set from longline fishing catch records across a 55 year period, using catch-per-unit-effort (# of large pelagic fish caught / # of "hooks" set with bait fish) as an estimate of predation rate. The CPUE is then estimated as a median value for a five year period in each 5x5 degree grid cell of ocean to test latitudinal pattern of CPUE as proxy for predation rate, and subsequently compared to independent estimate of species richness, for each of several ocean basins.

The results of these analyses do not support a higher CPUE rate in the tropics compared to temperate latitudes, and they show a negative relationship between this measure of predation rate and species richness of pelagic fish, contrary to prediction.

The approach is novel and of general (high) interest, using an impressive data set on longline fishing records. The manuscript is well-written and presentation of results is very clear. There are however several assumptions in the data and analyses, which require further consideration and scrutiny, before I would find the results robust and convincing. I suggest the following points be addressed:

(7) 1. As the authors point out, the data are reported catch of large pelagic fish on bait. Are there regional differences in bait or other methods that affect probability of catch? In some ways, disappearance of bait is equivalent to a "predation" event than catch per se. In many ways, the data are similar to those collected in "tethering experiments" that use live or dead prey, and either examine prey loss or predator catch, and there's always a question about how well these experiments estimate (approximate) actual predator-prey behavior, including a rich literature on this debate. I understand that catch data is what is available, but there should be some further discussion about the potential for bias and implications of both issues in interpretation (use) of these data as proxy for predation intensity.

This comment touches on similar issues that were also raised by Reviewer 1. We would thus like to refer to our detailed response to the reviewer comment 1, and particularly to the subsections (i) and (ii).

In direct response to the concern of tethering biases, we now directly address this literature, including a potentially rationale for why we think our main results would be less likely to be impacted by some overestimation of predation (lines 168-174). We further address the influence methodological variation in adding noise/bias to our results through extensive new analyses (see section starting at line 105), but we also acknowledge that we are unable to directly control for all factors adding variation into the available catch-per-unit-effort data, including factors such as bait type, soak time, and depth of hooks. Importantly, however, there is little *a priori* reason to believe that these factors vary systematically across latitude (see lines 115-119).

As well, we have revised our manuscript so that we are now very explicit what exactly our metric of 'relative predation' quantifies (lines 175-179), and how this metric is calculated (lines 62-66, 164-166). By using the term 'relative predation' we make sure that it is clear that what we measure is not total predation, but predation by large fish predators in the open ocean (which are the major predators in the bulk of our study area; see lines 179-181). We note that our metric of relative predation follows the rationale of how predation strength was quantified in several previous studies attempting to quantify predation (lines 166-168).

Finally, we would like to note that we consider it to be a strength of our study that the prey placed into the wild to measure predation constitutes real prey, and not just prey models as it is often the case in experiments.

(8) 2. Since they use CPUE as a proxy, what is known about the quality of reporting? I can imagine that only certain types of fish may be of interest (to those fishing) and others potentially discarded as bycatch. To consider “catch” as a measure of predation intensity, it seems important to include all species of fish caught (= total predation) versus only those of commercial value. I don't know much about this industry and whether there is a substantial portion of the catch this is not recorded. This should be addressed, to understand the data quality.

This comment is again much related to the first comment by Reviewer 1. Therefore, we would like to refer to the respective responses above (mainly subsections (i) and (ii)).

In short: We have added a new analyses in which we show that the resolution of reported predator taxa is not higher at those temperate latitudes where relative predation is strongest (Fig. S6). This rules out the possibility that stronger relative predation is caused by different reporting of predators across latitude.

We have now calculated taxon-specific patterns of relative predation across latitude, which we show as Figs S9 & S10. Importantly, the analysis of taxon-specific peaks of relative predation – irrespective of the frequency of a taxon in the total catch – reveals that most of these peaks fall on temperate latitudes and not near the equator (Fig. S8). We have also executed further GAMM-based analyses on predator taxa that make up a minor portion of the total catch by longline fisheries, and that are thus

deemed 'non-target' species. In line with our overall result, this analysis fails to find relative predation to be strongest near the equator (Fig. S7).

Finally, we would like to point to our extensive data filtering and pre-processing (see Methods) and to our test where we look for a possible influence of errors in reporting (i.e., high or low outlier values in the catch-per-unit-effort data) on latitudinal patterns of relative predation (Fig. S13).

Overall, we conclude that our general results of stronger predation exerted by large pelagic fish predators at temperate latitudes than near the equator are robust.

(9) 3. I have some concerns and questions about the effect of temporal variation on latitudinal pattern. The authors do a good job of exploring temporal pattern across 5 and 10 year time scales (Fig 2 and S10) to show consistent data at this time block, but I wonder whether the latitudinal pattern observed could be an artefact of pooling into five-year time bins and then looking at median values. I would expect considerable temporal variation and that this varies across latitude. In particular, in some marine systems (eg., benthic communities), predation can be very intense in temperate latitudes during the summer months, when it can be as high as or even higher than tropical latitudes. However, it may be that predation is more consistent in tropical systems for much of the year. If there is strong seasonality in temperature vs tropical latitudes, when predation is measured could greatly affect the latitudinal pattern observed (e.g., see discussion on recent paper by Cheng, Diversity & Distributions 2019, vol. 25). In this study, I would like to know whether there are seasonal differences in the catch data, such that (a) only certain seasons are represented in some latitudes (x ocean) and (b) whether season of measurement effects the results. For example, could higher catch rates in the temperate zone be the result of fishing only during the “high season” compared to tropical areas? Median values or other metrics would be affected by this.

It appears that the authors have the catch data by month, so I believe there are opportunities to test whether there are seasonal differences in both when data are collected and catch rates across latitude. While I don't mean that every aspect of this must be explored, it would provide more confidence to know that we aren't just seeing a difference in seasonality in effort or catch, and I can imagine various ways to approach this; one might be to compare summer and winter seasons separately to test whether results are robust to seasonality.

Reviewer 2 raises a good point about the potential for there to be seasonal differences in fishing efforts and catch rates. We did several things to address the potential for seasonal effects to be influencing the results:

(i) We fit new GAMMs where 'month' was set as fixed effect in addition to 'latitude' and 'time'. (We refrained from splitting the datasets into two or four seasons, as it is difficult to judge how exactly to split the data.) That is, for every month in every grid cell, we calculated median relative predation for each five-year period. The GAMMs revealed a very minor effect of 'month' (season) on relative predation, and the partial effect of latitude on CPUE remained qualitatively unchanged when compared to the main

analyses (Fig. S4). This suggests that seasonal aspects of fisheries do not drive our results.

(ii) To address the issue of effort variation across seasons (i.e., across months), we now plot monthly change in effort across 12 months, and latitude. We find that the overall fishing effort is substantial across the entire year (Fig. S5a). Despite some variation in effort during some parts of the year across latitude, this variation does not appear to influence our overall results (see Fig. S4). We further note that since effort is included in CPUE calculation, most of this variation is accounted for and is unlikely to explain the overall latitudinal patterns. We also cite Cheng et al. for the idea of geographic variation in predation due to season (line 108).

(iii) Reviewer 2 raised the possibility that the way we calculated 'relative predation' could mask possible seasonal variation.

We evaluated whether our results depend on the choice of (i) taking the *mean* monthly CPUE to calculate annual CPUE per grid cell, instead of calculating annual CPUE by dividing the yearly *sum* of all catches by the yearly *sum* of all hooks set per grid cell. (ii) We also carried out analyses where we utilized the *mean* annual CPUE, instead of the *median* annual CPUE per grid cell, per five-year time interval. We detail these analyses in the supplemental material (lines 37-47; see Fig. S15). Here, we would like to again refer to our analyses involving 'intercept-based relative predation', which are based on monthly data points. We outline these analyses in the Supplementary Material (lines 48-68; Figs S17-S21).

Overall, we find that these different approaches detect similar latitudinal patterns in relative predation, thus supporting the same overall conclusion of higher relative predation at temperate latitudes than near the equator.

(10) 4. I am also struck by several patterns in looking at the spatial display of data in Fig 1 that pose a few further questions and recommendations:

(a) First, it appears that the patterns adjacent to the continents (e Pacific and w Pacific) does display the predicted latitudinal pattern in the in the northern hemisphere. This is lost/swamped when combining with mid ocean. Is there a difference between continental shelf/adjacent and mid ocean, or with distance from shore x latitude? If so, this is interesting and worth discussion. More broadly, it would be good to discuss briefly what are the expectation (and past literature) for open ocean pelagic fish?

We agree that a continental-only analysis would be valuable. However, because our data are aggregated in 5°x5° grids (this is the grain at which most of the fishery data is available due to protection of privacy and detailed movements of individual fleets), we do not have enough replication/power to carry out an analysis for grids whose midpoint is 100 km or less from shore (this is about the average extent of the continental shelf). Part of the problem is that certain latitudes lack sufficient coastline to make this analysis viable. Moreover, even when a 5°x5° grid cell is located right next to a continent, it will include substantial data that was not collected within the continental shelf.

However, we have now carried out new GAMMs in which we included the effect of 'distance to land' and 'ocean depth' – two variables that should account for continental effects (such as possible influences of the continental shelf) (Fig S14B). Although we find that some variation in relative predation is explained by these additional spatial variables, the latitudinal pattern remains largely unchanged when accounting for this variation. We now communicate this finding in the main text (lines 113-115): *"Similarly, we found our results to be robust to analyses accounting for variation related to other spatial variables (longitude, proximity to land and ocean depth; Fig. S14)."*

We also discuss the possibility of underestimating predation in temperate regions (Supplementary Materials lines 182-185): *"We further note that several pelagic fish predators largely considered by-catch of logline fisheries (such as many shark species) are rich at temperate latitudes⁴, raising the possibility that we may even underestimate the degree to which predation is greater at temperate latitudes than near the equator."*

(11) (b) Second, I'm curious why the eastern and western Atlantic aren't separated in the analysis in parallel to both sides of the Pacific. It seems like they should be and would show a different pattern than the entire Atlantic. If this is an issue with space, it could be added to supplemental.

Indeed, this was not all that clear before! We now clarify that the reason why the Pacific was analyzed in two separate analyses is because the longline fisheries datasets for the West and East Pacific are curated by different commissions. Consequently, the datasets differ in some aspects (see Table S2). We now make this clear in the legend to Table S1 (*"Because these datasets are curated by different commissions and differ in some aspects (see below), we analyzed these datasets independently."*), as well as in the Methods (lines 141-144: *"We quantified the relative predation that is exerted by large pelagic fish predators based on four publicly available datasets from pelagic longline fishing (West Pacific, East Pacific, Atlantic and Indian Ocean), each managed by an independent commission (see Table S1 for details)."*)

We hope Reviewer 2 agrees that there is little *a priori* biological reason for splitting other oceans (which are single datasets) into multiple separate analyses.

(12) (c) Third, it's also interesting how much higher the catch appears in the southern ocean, and how much lower in the NE Atlantic. Does this suggest historical effects (or lack) of fishing pressure, or is this driven by productivity, or some combination. While exploration of this is beyond the scope of the current paper, it would be worth some mention of possible drivers for this (and especially the southern ocean, since this is such a prominent feature) --- in just a sentence.

Indeed, this difference between the hemispheres is interesting. We agree that stronger fishing pressure and thus greater oceanic exploitation in the Northern hemisphere may contribute to why baseline relative predation is generally higher in the Southern hemisphere; however, the same trend is already evident in the 60's when there was still relatively little fishing and oceanic exploitation overall. We now state this reasoning in

the main text (lines 98-101): "Similarly, the stronger fishing pressure and thus greater oceanic exploitation in the Northern hemisphere⁴² is unlikely to be the sole explanation of why relative predation is generally higher in the Southern hemisphere because this difference between hemispheres was already evident in the early 60's (Fig. 2)."

We hope Reviewer 2 agrees with our preference to not speculate further what else could drive this pattern.

Response Letter References

1. Roslin T, *et al.* Higher predation risk for insect prey at low latitudes and elevations. *Science* **356**, 742-744 (2017).
2. McKinnon L, *et al.* Lower predation risk for migratory birds at high latitudes. *Science* **327**, 326-327 (2010).
3. Hargreaves AL, *et al.* Seed predation increases from the Arctic to the Equator and from high to low elevations. *Science Advances* **5**, eaau4403 (2019).
4. Lövei GL, Ferrante M. A review of the sentinel prey method as a way of quantifying invertebrate predation under field conditions. *Insect science* **24**, 528-542 (2017).
5. Falkowski PG, Wilson C. Phytoplankton productivity in the North Pacific ocean since 1900 and implications for absorption of anthropogenic CO₂. *Nature* **358**, 741 (1992).
6. Rykaczewski RR, Dunne JP. A measured look at ocean chlorophyll trends. *Nature* **472**, E5 (2011).
7. Barneche DR, *et al.* Feeding macroecology of territorial damselfishes (Perciformes: Pomacentridae). *Marine Biology* **156**, 289-299 (2009).
8. Wilson AB. Fecundity selection predicts Bergmann's rule in syngnathid fishes. *Mol Ecol* **18**, 1263-1272 (2009).
9. Perez KO, Munch SB. Extreme selection on size in the early lives of fish. *Evolution: International Journal of Organic Evolution* **64**, 2450-2457 (2010).
10. Macpherson E, Duarte CM. Patterns in species richness, size, and latitudinal range of East Atlantic fishes. *Ecography* **17**, 242-248 (1994).
11. Smith KF, Brown JH. Patterns of diversity, depth range and body size among pelagic fishes along a gradient of depth. *Glob Ecol Biogeogr* **11**, 313-322 (2002).

12. Fisher JA, Frank KT, Leggett WC. Breaking Bergmann's rule: truncation of Northwest Atlantic marine fish body sizes. *Ecology* **91**, 2499-2505 (2010).
13. Fisher JA, Frank KT, Leggett WC. Global variation in marine fish body size and its role in biodiversity–ecosystem functioning. *Marine Ecology Progress Series* **405**, 1-13 (2010).
14. Schemske DW, Mittelbach GG, Cornell HV, Sobel JM, Roy K. Is there a latitudinal gradient in the importance of biotic interactions? *Annu Rev Ecol Evol Syst* **40**, 245-269 (2009).
15. Mittelbach GG, *et al.* Evolution and the latitudinal diversity gradient: speciation, extinction and biogeography. *Ecol Lett* **10**, 315-331 (2007).
16. Rohde K. Latitudinal gradients in species diversity: the search for the primary cause. *Oikos*, 514-527 (1992).
17. Schemske DW, Mittelbach GG. "Latitudinal Gradients in Species Diversity": Reflections on Pianka's 1966 Article and a Look Forward. *Am Nat* **189**, 599-603 (2017).

Supporting analysis

A

B

GAMM with 'time', 'latitude' and 'chlorophyll-a' as explanatory variables of attack rate

Method

We obtained remotely-sensed estimates of sea-surface chlorophyll-a concentrations (in mg/m^3) between 1997 and 2010 from NOAA. Because no such data are available for the time prior to 1997, we calculated the average chlorophyll-a concentration per unique latitude x longitude combination based on all available data. We further averaged these productivity estimates within $\pm 2^\circ$ of latitude and longitude to obtain a productivity estimate for most geographic locations for which longline catch statistics were available ($> 99\%$ of the longline data). Notably, the correlation between time-specific (considering both month and year) and time-averaged productivity estimates across those longline data entries with an available time-specific productivity estimate was high (average Pearson's r of the four data sets was 0.73), indicating that the temporal variability in chlorophyll-a concentration is relatively small compared to the spatial one. The global map shown in panel (A) depicts median chlorophyll-a concentration.

We then ran GAMMs in which we added 'chlorophyll-a' as another fixed effect besides 'latitude' and 'time interval'. The partial effect of latitude on relative attack rate from these models is shown as panel (B).

Result:

In this analysis we re-calculated latitudinal patterns of relative predation while accounting for variation in 'chlorophyll-a' concentration – a proxy for water turbidity (following e.g., refs. ^{5, 6}). The latitudinal patterns of relative fish predation look very similar to the ones obtained without accounting for water turbidity (compare to Fig. 1B).

'Chlorophyll-a' data source link: <https://coastwatch.pfeg.noaa.gov/erddap/files/erdSWchla8day/>

Reviewers' Comments:

Reviewer #1:

Remarks to the Author:

I am very impressed at the thoroughness and detail of the authors' responses to the referees' criticisms. These allay my initial concerns and I recommend publication with minor revision.

Title

Perhaps it should say "Pelagic fish predation is stronger at temperate latitudes than the equator" ("worlds open ocean" is more wordy and ambiguous)

Abstract

It's odd that some oceanographers use the word 'biome' in this way - the term was originally and is now well-established in terrestrial ecology for large areas typified by plants that provide habitat for other species (marine parallels would be mangroves, kelp forests); open water phytoplankton do not really provide the same functional structure. I recommend dropping its use in this context to not perpetuate confusion.

Introduction

Line

42 - see alternative model proposed and based on pelagic foraminifera by Brayard, A., Escarguel, G. and Bucher, H., 2005. Latitudinal gradient of taxonomic richness: combined outcome of temperature and geographic mid-domains effects?. *Journal of Zoological Systematics and Evolutionary Research*, 43(3), pp.178-188.

55 - I do not follow how predation is "minimally related to latitude in herbivory" as implied here or what is the meaning (missing words?)

Results

125 - note Aquamaps overestimates richness because it is range maps (error of commission) but observations have errors of omission. I think worth mentioning this because if species abundance has declined near the equator (such as due to high temperatures) then this may drive the latitudinal gradient to be bimodal (as in Chaudhary et al and other papers and in this paper). Observations from recent decades may show this. However, range maps based on longer-term data may not (because a fish may once have been in the equatorial region).

132 - I'd be cautious with the metric of speciation rates as it is obtuse and may be incorrect. It seems contrary to other empirical data on richness and endemism.

I'd like to see the species data in the main MS. The SM is optional reading but I think it is essential to have some insights into what exactly has been studied here.

Reviewer #2:

Remarks to the Author:

Overall, the manuscript is of high quality and broad interest. It has been improved by revisions, which serve to both clarify and demonstrate robustness of the patterns at the scale of ocean basins. The authors included several additional analyses in the Supplemental Materials that are helpful in this regard.

There are two issues that could be further clarified, including one from the initial reviewer comments that focuses on seasonal variation. The question involves the effect of season on the latitudinal pattern

of relative predation rate by basin. The authors have added the effect of month to the GAMM analyses, to test the partial effects of latitude and month on relative predation, including two figures (S4 and S5) in Supplemental Materials. The analyses show that on average there is no (low) effect of season, but it appears that month is treated as a fixed term and does not adjust for seasonal differences between hemispheres (which are six months out of phase). Since these are out of phase, can the disparate effects of hemisphere simply cancel out on average across the analysis? The plot for Fig S4A is informative, as it shows the monthly variation (dots) by latitude. The same is true for Fig 5B, which shows monthly variation in Effort by ocean basin. I think it would be of broad interest to see a plot of relative predation by month across latitude and ocean basin --- similar to Fig 5B, or it could be a panel of month by basin. This is another way to address how seasonally variable the latitudinal pattern of predation is by month --- and whether there are some months/seasons when there is a strong latitudinal pattern. I would find this type of presentation a valuable addition, and I don't think it would be a big lift to include it.

The other issue involves the temporal stability of latitudinal pattern across deeper time. The presentation of decadal changes in relative predation is very nice, indicating both that relative predation is changing and it has not demonstrated the expected latitudinal pattern (of higher tropical vs temperate rates) at any time since 1960. These are robust and compelling results. However, I am not convinced that human fishing pressure may have strongly influence the observed pattern, especially between northern and southern hemisphere as suggested on lines 98-104. I believe there is a strong case to be made for significant human effects on fishery resources over centuries and earlier, especially in the northern hemisphere (and northern Atlantic in particular, where CPUE is especially low). I recommend a bit more caution in this interpretation, including the use of the term "resilient" (which doesn't seem to quite apply here).

I have two other minor comments. First, the text on lines 53-55 seems a little confusing regarding predation on bird eggs and nests (stronger in tropics) versus bird nests (minimal pattern). Maybe these could be combined into a single statement? Second, one line 141, I think the a word is missing and "by" should be added to read ..."exerted by large"...

Dear Reviewers,

Thank you again for your thoughtful comments! Below, we explain in detail how we addressed each one of them. Revising our manuscript accordingly has resulted in one further supplementary analysis, a new main figure (moved from the SupplMat), as well as additions to, and clarifications of the text. We hope you agree that these revisions have further strengthened our paper, which we now hope is ready for publication.

Best wishes,
Marius Roesti, Roi Holzman & Co-Authors

Reviewers' comments

Reviewer #1 (Remarks to the Author):

I am very impressed at the thoroughness and detail of the authors' responses to the referees' criticisms. These allay my initial concerns and I recommend publication with minor revision.

Reviewer comment 1

Title

Perhaps it should say "Pelagic fish predation is stronger at temperate latitudes than the equator" ("worlds open ocean" is more wordy and ambiguous)

>> This is a good suggestion, and we have decided to change the manuscripts' title accordingly.

Reviewer comment 2

Abstract

It's odd that some oceanographers use the word 'biome' in this way - the term was originally and is now well-established in terrestrial ecology for large areas typified by plants that provide habitat for other species (marine parallels would be mangroves, kelp forests); open water phytoplankton do not really provide the same functional structure. I recommend dropping its use in this context to not perpetuate confusion.

>> We have followed this suggestion and no longer use the 'biome' terminology. See line 23 in the abstract.

Introduction

Line

Reviewer comment 3

42 - see alternative model proposed and based on pelagic foraminifera by Brayard, A., Escarguel, G. and Bucher, H., 2005. Latitudinal gradient of taxonomic richness: combined outcome of temperature and geographic mid-domains effects?. Journal of Zoological Systematics and Evolutionary Research, 43(3), pp.178-188.

>> Thank you for this reference suggestion. We now cite Brayard et al. on line 42, providing another dissenting view to the main hypotheses in the field.

Reviewer comment 4

55 - I do not follow how predation is "minimally related to latitude in herbivory" as implied here or what is the meaning (missing words?)

>> We have clarified this passage, and particularly the wording on the consumption target (line 52-60)

Reviewer comment 5

Results

125 - note Aquamaps overestimates richness because it is range maps (error of commission) but observations have errors of omission. I think worth mentioning this because if species abundance has declined near the equator (such as due to high temperatures) then this may drive the latitudinal gradient to be bimodal (as in Chaudhary et al and other papers and in this paper). Observations from recent decades may show this. However, range maps based on longer-term data may not (because a fish may once have been in the equatorial region).

>> While *AquaMaps* is indeed likely to contain some errors, the pattern we derive from the *AquaMaps* data approximates the common expectation of greater species richness at lower latitudes. Moreover, there is little reason to believe that errors – such as errors of (c)omission – would vary systematically across latitude. Hence we believe the richness estimates provided by *AquaMaps* are of appropriate quality for the purpose of our study. However, we now make note of the reviewer's concerns in lines (310-314):

"We note that the species richness estimated by AquaMaps⁴⁶ is based on range maps, potentially leading to richness overestimates due to errors of commission. However, the latitudinal species richness pattern we observe is likely to be robust to this error because this and related reporting errors are unlikely to vary systematically across latitude."

We further note that we are interested in the long-term diversity pattern across latitude because this is the pattern that is most likely influenced by variation in the strength of biotic interactions as estimated in our study.

Reviewer comment 6

132 - I'd be cautious with the metric of speciation rates as it is obtuse and may be incorrect. It seems contrary to other empirical data on richness and endemism.

>> We believe that discussing the exact methodology Rabosky et al. used, and its potential issues, is beyond the scope of this paper. Given the concerns by Reviewer 1, we have altered the wording to express more doubt regarding the results by Rabosky and colleagues. Please see lines 139-143.

Reviewer comment 7

I'd like to see the species data in the main MS. The SM is optional reading but I think it is essential to have some insights into what exactly has been studied here.

>> We have moved the figure depicting density of taxon-specific predator peaks across latitude to the main manuscript (see the new main Fig. 3). In the respective figure legend, we now indicate the exact number of predator taxa (i.e., species or groups of related species) investigated within each ocean basin, while referring to the extensive Supplementary Table 3 for further details. Moreover, we include an explanation of this new main figure in lines 119-122.

Reviewer #2 (Remarks to the Author):

Overall, the manuscript is of high quality and broad interest. It has been improved by revisions, which serve to both clarify and demonstrate robustness of the patterns at the scale of ocean basins. The authors included several additional analyses in the Supplemental Materials that are helpful in this regard.

Reviewer comment 8

There are two issues that could be further clarified, including one from the initial reviewer comments that focuses on seasonal variation. The question involves the effect of season on the latitudinal pattern of relative predation rate by basin. The authors have added the effect of month to the GAMM analyses, to test the partial effects of latitude and month on relative predation, including two figures (S4 and S5) in Supplemental Materials. The analyses show that on average there is no (low) effect of season, but it appears that month is treated as a fixed term and does not adjust for seasonal differences between hemispheres (which are six months out of phase).

Since these are out of phase, can the disparate effects of hemisphere simply cancel out on average across the analysis? The plot for Fig S4A informative, as it shows the monthly variation (dots) by latitude. The same is true for Fig 5B, which shows monthly variation in Effort by ocean basin. I think it would be of broad interest to see a plot of relative predation by month across latitude and ocean basin --- similar to Fig 5B, or it could be a panel of month by basin. This is another way to address how seasonally variable the latitudinal pattern of predation is by month --- and whether there are some months/seasons when there is a strong latitudinal pattern. I would find this type of presentation a valuable addition, and I don't think it would be a big lift to include it.

>> Reviewer 2 makes a good point here. We have followed her/his suggestion and now include an analysis into the Supplementary Material in which we show relative predation per month across latitude (Figure S6). Although we generally find there to be more seasonal (monthly) variation in relative predation at higher latitudes, the mean across all monthly estimates per latitude supports the general pattern of highest relative predation away from the equator. We hope Reviewer 2 agrees that further exploration of seasonal variation is beyond the scope of this study paper.

Reviewer comment 9

The other issue involves the temporal stability of latitudinal pattern across deeper time. The presentation of decadal changes in relative predation is very nice, indicating both that relative predation is changing and it has not demonstrated the expected latitudinal pattern (of higher tropical vs temperate rates) at any time since 1960. These are robust and compelling results. However, I am not convinced that human fishing pressure may have strongly influence the observed pattern, especially between northern and southern hemisphere as suggested on lines 98-104. I believe there is a strong case to be made for significant human effects on fishery resources over centuries and earlier, especially in the northern hemisphere (and northern Atlantic in particular, where CPUE is especially low). I recommend a bit more caution in this interpretation, including the use of the term "resilient" (which doesn't seem to quite apply here).

>> We have rephrased this passage to be more cautious about this interpretation (notably, we have removed 'resilient' altogether). Please see lines 107-110.

Reviewer comment 10

I have two other minor comments. First, the text on lines 53-55 seems a little confusing regarding predation on bird eggs and nests (stronger in tropics) versus bird nests (minimal pattern). Maybe these could be combined into a single statement?

>> This suggestion is in line with comment 4 by Reviewer 1. We have now clarified this text passage.

Reviewer comment 11

Second, one line 141, I think the a word is missing and "by" should be added to read ..."exerted by large"...

>> Typo fixed!

Reviewers' Comments:

Reviewer #2:

Remarks to the Author:

The manuscript was initially of high quality and has been further improved by additions in revision. I appreciate the additional analyses and responsiveness by authors, to explore/test/present additional dimensions of this interesting and impressive data set.

I think the manuscript is of broad interest and recommend publication. It is well-written and provides an in-depth and extensive analysis, which I expect will have considerable impact and stimulate further discussion and work on the topic.

I do have some additional further comments and recommendations for consideration by the authors and editor(s) before publication. All of these are relatively easy text changes, mostly to clarify presentation or interpretation. I don't feel that I need to see a revised manuscript, and outline specific comments and recommendations for minor revision below, and many of these focus on the most recent additions/revisions highlighted by the authors. The most substantive comments involve lines 107-110 and 113-118.

- Title: I agree that this is an improved title that better reflects content.
- Lines 52-59 highlighted text: This could be smoothed a bit further. I suggest revising this text to read something like: "For example, some studies report that predation on bird eggs and nestlings, insects, marine invertebrates, and seeds increases in strength towards the equator. Other studies found no relationship in strength of bird nest predation or marine herbivory with latitude, and some show inverse patterns, including stronger predation on brachiopods at temperate versus tropical latitudes."
- Line 61: I suggest replacing "typical" with "often", as I believe we are seeing a wave of recent papers that are not at small scales.
- Line 67: I suggest replacing the second use of "relative" with another term, since it is already used a few words back. Maybe could say "... relative predation across latitude".
- Line 92: I suggest omitting "open" from the term "open ocean", since some sites are coastal --- and "open" implies some distance (undefined) from land. I think just using "ocean" is also consistent with usage elsewhere in manuscript.
- Lines 107-110: I don't believe that any strong statement can be made about "baseline patterns for predation" across space using fishing data/catch from 1960s onward. This is very recent in time, and this statement or framing (and that from the previous sentence) imply otherwise, implying extrapolation much further back (timescale undefined) and possibly to evolutionary time scales. Yet, there is extensive evidence of fisheries depletion, including ocean fishing, back centuries to millennia -- especially in the northern hemisphere. I don't know if the authors intend this or not, but the inference is a red flag and should be changed. I don't believe limiting the inference to recent time is a weakness, as the pattern is strong and convincing. In fact, I find making a broader claim (and extending this back deeper in time) is a distraction that detracts from the story.
- Lines 113-118 & Figure S6: This section reads fine overall but omits any mention of Fig S6, which was added at reviewer (my) suggestion. I don't see any mention of Fig S6 in text. I appreciate this being added to Supplementary Material, as it adds an important dimension to the data interpretation -

-- at least from my perspective. While it supports the overall pattern of predation across latitude, it shows the seasonal variation and that latitudinal pattern is driven by summer peaks at temperate latitude. I don't think a lot needs to be said, but suggest that some reference / statement be made to this effect in main text, possibly by inserting a brief additional bullet after (i) with a sentence of portion of sentence to make some such statement.

- Lines 119-122: Suggest minor word change to: "For example, when examining individual taxa, most peaks in predation do not fall..."
- Line 125: Seems like both "a priori" and your extensive analysis support this. I suggest adding the latter, saying something like "... we have no evidence or a priori reason to believe..."
- Line 130: I think you could omit "probably" here.
- Line 136: change "at the equator" to "near the equator", since many peaks are not exactly here.
- Line 136: add "many" to "other groups" to read "many other groups" --- since this also is not universally so.
- Line 143: I suggest removing "now charged" as this sets an odd tone for the last sentence. The sentence also seems a bit wordy and could be more crisply stated, to underscore the need to explain this pattern.
- Line 179: reference "543" should be "54".
- Fig S6, line 175: I think the caption reads fine as is, but could replace "appears to be strong" with "are stronger", depending on whether journal requires a statistic to support this. I think this is very evident from the plots, that the Coefficient of Variation increases at high latitude.

RESPONSE LETTER TO REVIEWER COMMENTS

Dear Reviewer #2,

Thank you for your last set of comments. Below, we explain in detail how we addressed each one of them.

Best wishes,
Marius Roesti, Roi Holzman & Co-Authors

#####

Reviewer #2

The manuscript was initially of high quality and has been further improved by additions in revision. I appreciate the additional analyses and responsiveness by authors, to explore/test/present additional dimensions of this interesting and impressive data set.

I think the manuscript is of broad interest and recommend publication. It is well-written and provides an in-depth and extensive analysis, which I expect will have considerable impact and stimulate further discussion and work on the topic.

I do have some additional further comments and recommendations for consideration by the authors and editor(s) before publication. All of these are relatively easy text changes, mostly to clarify presentation or interpretation. I don't feel that I need to see a revised manuscript, and outline specific comments and recommendations for minor revision below, and many of these focus on the most recent additions/revisions highlighted by the authors. The most substantive comments involve lines 107-110 and 113-118.

Comment 1

- Title: I agree that this is an improved title that better reflects content.
- >> Thank you.

Comment 2

- Lines 52-59 highlighted text: This could be smoothed a bit further. I suggest revising this text to read something like: "For example, some studies report that predation on bird eggs and nestlings, insects, marine invertebrates, and seeds increases in strength towards the equator. Other studies found no relationship in strength of bird nest predation or marine herbivory with latitude, and some show inverse patterns, including stronger predation on brachiopods at temperate versus tropical latitudes."
- >> We have changed this passage accordingly. We think it now reads very smoothly and is clear while highlighting the different results from these studies.

Comment 3

- Line 61: I suggest replacing "typical" with "often", as I believe we are seeing a wave of recent papers that are not at small scales.
- >> We have replaced 'typical' by 'often'.

Comment 4

- Line 67: I suggest replacing the second use of "relative" with another term, since it is already used a few words back. Maybe could say "... relative predation across latitude".

>> This wording no longer applies since we have slightly changed this sentence in order to address the Editorial request to summarize our results at the end of the Introduction.

Comment 5

- Line 92: I suggest omitting “open” from the term “open ocean”, since some sites are coastal --- and “open” implies some distance (undefined) from land. I think just using “ocean” is also consistent with usage elsewhere in manuscript.

>> Note that the pelagic zone makes up a substantial portion of the open ocean, and the part of the open ocean that is studied most. Importantly, our measure of fish predation is based on *pelagic* longline fisheries data only. We thus want to make sure it is clear that our inference of predation strength does not quantify, for example, fish predation in coral reefs. We therefore prefer keeping it as 'open ocean'.

Comment 6

- Lines 107-110: I don't believe that any strong statement can be made about “baseline patterns for predation” across space using fishing data/catch from 1960s onward. This is very recent in time, and this statement or framing (and that from the previous sentence) imply otherwise, implying extrapolation much further back (timescale undefined) and possibly to evolutionary time scales. Yet, there is extensive evidence of fisheries depletion, including ocean fishing, back centuries to millennia --- especially in the northern hemisphere. I don't know if the authors intend this or not, but the inference is a red flag and should be changed. I don't believe limiting the inference to recent time is a weakness, as the pattern is strong and convincing. In fact, I find making a broader claim (and extending this back deeper in time) is a distraction that detracts from the story.

>> We agree with Reviewer 2 that we are, and will always be, unable to know the pattern before any human impact. However, this is true for any other study that experimentally tests latitudinal patterns of biotic interactions because humans have impacted the globe to some degree way before any such experiments were conducted (both on land and in the seas).

We also agree with Reviewer 2 that we are unable to know whether detected patterns in recent time are mirroring patterns in deep evolutionary time (which is again true for any study). Notably, this is not only because of the relatively recent appearance of humans on the planet, but also because some environmental conditions have changed naturally on our planet during deeper evolutionary time.

To address the concerns by Reviewer 2, we now make very explicit that we are unable to know the pattern before any human impact on the ocean, nor in deep evolutionary time. Nevertheless, it is a clear benefit and novel aspect of our study that we can look for consistency in the latitudinal pattern across decades, suggesting that the qualitative latitudinal pattern has been stable over this time period. To the best of our knowledge, this is the first study testing this hypothesis that has included a temporal dimension.

This section now reads like this: "Although we are unable to infer patterns of pelagic fish predation in deep evolutionary time or prior to any human impact on the ocean, the relative stability of the observed latitudinal pattern in more recent time – despite an increase in human fishing pressure – suggests that peaks in pelagic fish predation away from the equator may have been the prevailing pattern over time."

Comment 7

- Lines 113-118 & Figure S6: This section reads fine overall but omits any mention of Fig S6, which was added at reviewer (my) suggestion. I don't see any mention of Fig S6 in text. I appreciate this being added to Supplementary Material, as it adds an important dimension to the data interpretation --- at least from my perspective. While it supports the overall pattern of predation across latitude, it shows the seasonal variation and that latitudinal pattern is driven by summer peaks at temperate latitude. I don't think a lot needs to be said, but suggest that some reference / statement be made to this effect in main text, possibly by

inserting a brief additional bullet after (i) with a sentence of portion of sentence to make some such statement.

>> Following the Reviewer's suggestion, we have added an explicit statement concerning variation in relative predation pattern across latitude as an additional bullet after (i).

Comment 8

• Lines 119-122: Suggest minor word change to: "For example, when examining individual taxa, most peaks in predation do not fall..."

>> We have optimized wording of this sentence. However, we prefer to not start this sentence with 'For example,' as suggested by the reviewer, because we here describe an additional analysis and not an example for anything said previously.

This sentence now reads like this: "We further examined predation peaks of individual predator taxa and found that a majority of these peaks fall at absolute latitudes greater than 10°, and not near the equator (Figure 3)."

Comment 9

• Line 125: Seems like both "a priori" and your extensive analysis support this. I suggest adding the latter, saying something like "... we have no evidence or a priori reason to believe..."

>> We have changed this sentence, it now reads like this: " Thus, although some variation in relative predation is likely to be explained by how and when the underlying data were collected, we found no evidence or have little *a priori* reason to believe (with regards of factors we were unable to directly address, such as soak time, hook depth, and bait type) that this variation does more than add noise to what is otherwise a robust latitudinal pattern in the strength of predation exerted by large pelagic fish."

Comment 10

• Line 130: I think you could omit "probably" here.

>> We agree that 'probably' sounds a bit odd. We have decided to delete this part of the sentence altogether.

Comment 11

• Line 136: change "at the equator" to "near the equator", since many peaks are not exactly here.

>> Well spotted. We have changed the wording as suggested

Comment 12

• Line 136: add "many" to "other groups" to read "many other groups" --- since this also is not universally so.

>> We think there is not enough evidence to support the notion that this is true for 'many' other groups, but that there is evidence for this pattern in 'some' other groups. Thus, we have changed the wording from 'other groups' to 'some other groups'.

Comment 13

• Line 143: I suggest removing "now charged" as this sets an odd tone for the last sentence. The sentence also seems a bit wordy and could be more crisply stated, to underscore the need to explain this pattern.

>> We agree the sentence should be crisper and have shorted it. However, we have chosen to retain the "now charged" wording since we intend to make this call to action for further work and consideration of this issue as both valid and worthy of further research. The last sentence of the main paper thus now reads like this: "Evolutionary ecologists are now charged with explaining why the eco-evolutionary processes thought to generate and maintain diversity are not always strongest in the most diverse regions on Earth."

Comment 14

- Line 179: reference “543” should be “54”.

>> We corrected this typo accordingly.

Comment 15

- Fig S6, line 175: I think the caption reads fine as is, but could replace “appears to be strong” with “are stronger”, depending on whether journal requires a statistic to support this. I think this is very evident from the plots, that the Coefficient of Variation increases at high latitude.

>> We have changed the wording as suggested.